# Clinicopathological Factors Associated with Oncotype DX Risk Group in Patients with ER+/HER2- Breast Cancer

**DOI:** 10.3390/cancers15184451

**Published:** 2023-09-07

**Authors:** Ran Song, Dong-Eun Lee, Eun-Gyeong Lee, Seeyoun Lee, Han-Sung Kang, Jai Hong Han, Keun Seok Lee, Sung Hoon Sim, Heejung Chae, Youngmee Kwon, Jaeyeon Woo, So-Youn Jung

**Affiliations:** 1Department of Surgery, Center of Breast Cancer, National Cancer Center, Goyang 10408, Republic of Korea; thdfks37@hanmail.net (R.S.); jaeyeon1205@ncc.re.kr (J.W.); 2Biostatistics Collaboration Team, Research Core Center, Research Institute of National Cancer Center, Goyang 10408, Republic of Korea; 3Department of Medical Oncology, Center of Breast Cancer, National Cancer Center, Goyang 10408, Republic of Korea; 4Department of Pathology, Center of Breast Cancer, National Cancer Center, Goyang 10408, Republic of Korea

**Keywords:** breast cancer, Oncotype DX, recurrence score, prediction, prognosis

## Abstract

**Simple Summary:**

Oncotype DX (ODX) is a gene assay that can predict the recurrence risk in patients with early breast cancer. We aimed to evaluate clinicopathological factors that can predict the ODX risk group as an alternative to this expensive assay. A total of 547 patients with estrogen receptor-positive, human epidermal growth factor 2-negative, and lymph node-negative breast cancer who underwent ODX testing were retrospectively included in this study; these were categorized into risk groups based on their age and recurrence scores. We identified that the ODX risk groups were significantly associated with histologic grade, progesterone receptor expression, Ki-67, and p53 expression in patients aged ≤50 years and with the tumor size, Ki-67, and p53 expression in patients aged >50 years. These clinicopathological factors, which differ with age, can be used to predict the ODX risk group and decide on the need for adjuvant chemotherapy.

**Abstract:**

Oncotype DX (ODX), a 21-gene assay, predicts the recurrence risk in early breast cancer; however, it has high costs and long testing times. We aimed to identify clinicopathological factors that can predict the ODX risk group and serve as alternatives to the ODX test. This retrospective study included 547 estrogen receptor-positive, human epidermal growth factor receptor 2-negative, and lymph node-negative breast cancer patients who underwent ODX testing. Based on the recurrence scores, three ODX risk categories (low: 0–15, intermediate: 16–25, and high: 26–100) were established in patients aged ≤50 years (*n* = 379), whereas two ODX risk categories (low: 0–25 and high: 26–100) were established in patients aged >50 years (*n* = 168). Factors selected for analysis included body mass index, menopausal status, type of surgery, and pathological and immunohistochemical features. The ODX risk groups showed significant association with histologic grade (*p* = 0.0002), progesterone receptor expression (*p* < 0.0001), Ki-67 (*p* < 0.0001), and p53 expression (*p* = 0.023) in patients aged ≤50 years. In patients aged >50 years, tumor size (*p* = 0.022), Ki-67 (*p* = 0.001), and p53 expression (*p* = 0.001) were significantly associated with the risk group. Certain clinicopathological factors can predict the ODX risk group and enable decision-making on adjuvant chemotherapy; these factors differ according to age.

## 1. Introduction

Breast cancer is the most common cancer among women worldwide, and its annual incidence continues to increase in Korea [1,2]. Patients with early breast cancer have experienced survival benefits with adjuvant chemotherapy after curative resection [3,4]; however, some patients have reportedly experienced minimal benefits despite various degrees of toxicity [5,6]. Multigene assays can be performed to predict the risk of recurrence in these patients and thus reduce unnecessary administration of cytotoxic chemotherapy [7].

Oncotype DX (ODX) (Genomic Health, Redwood City, CA) is a genomic test that uses quantitative real-time reverse transcriptase–polymerase chain reaction for 16 cancer genes and 5 reference genes [8,9]. It can predict the recurrence risk and estimate the benefit of adjuvant chemotherapy in patients with estrogen receptor (ER)-positive, human epidermal growth factor receptor 2 (HER2)-negative early breast cancer [8,9,10]. Previous studies have shown that patients with a low recurrence score (RS) on the ODX have a low recurrence rate with endocrine therapy alone [11,12,13]. Therefore, current guidelines recommend that the ODX test be performed to decide on the need for the addition of adjuvant chemotherapy to endocrine therapy for these patients [14,15]. Although the ODX test provides prognostic information that allows decision-making on the need for adjuvant chemotherapy, ODX cannot be performed for all eligible patients because of its high cost and long testing time [16,17,18].

Therefore, identifying the clinicopathological factors associated with the ODX RS can guide decision-making on adjuvant treatment and reduce medical costs as an alternative to the expensive ODX test. This study aimed to evaluate the clinicopathological factors that can predict the ODX risk group in the Korean population. Unlike previous studies that developed nomograms for the entire study population, we clarified and visualized the results by constructing a decision tree using an age cutoff of 50 years.

## 2. Materials and Methods

### 2.1. Patient and Variable Selection

Patients with breast cancer who underwent curative resection and ODX testing at the National Cancer Center of Korea between October 2011 and December 2021 were retrospectively enrolled in this study. The inclusion criteria were as follows: (1) age ≥ 19 years, (2) histopathologically confirmed invasive breast cancer, (3) confirmation of an ER-positive and HER2-negative subtype based on immunohistochemistry or in situ hybridization test for HER2, and (4) pathological confirmation of lymph node (LN)-negative status. Male patients, as well as women who did not undergo axillary staging, were excluded.

Clinical and pathological data were retrieved from the patient’s electronic medical records. Clinical data included patient age, body mass index, menopausal status, and type of surgery. Data on the treatment status, including adjuvant chemotherapy, endocrine therapy, and radiation therapy, were also obtained. Pathological data included tumor size, multiplicity, histologic type, histologic grade, nuclear grade, lymphovascular invasion (LVI), and hormone receptor status, including ER, progesterone receptor (PR), and androgen receptor (AR), Ki-67, and p53 status. The ER, PR, and AR statuses were evaluated using an Allred score between 0 and 8. An Allred score ≤ 2 was considered negative, and a score > 2 was considered positive. Notably, the AR data have been recorded since 20 March 2016; therefore, data on the AR status were unavailable for 257 patients who underwent surgery prior to this date. Ki-67 was presented as a percentage, ranging from 0 to 100% in consecutive numbers. For the p53 status, staining under 10% was regarded as negative, and the rest was regarded as positive.

### 2.2. Risk Stratification by the ODX RS

The RS cutoff value for high risk was 25 based on findings from previous clinical trials, such as the Trial Assigning Individualized Options for Treatment (TAILORx) and the Rx for Positive Node, Endocrine Responsive Breast Cancer (RxPONDER) trial [11,19]. Using this RS cutoff value, patients aged >50 years were stratified into two risk groups (0–25, low-risk group and 26–100, high-risk group). Conversely, patients aged ≤50 years were stratified into three risk groups based on previous studies and current guidelines [13,14,20]. The RS cutoff values for low-risk and high-risk were 15 and 25, respectively (0–15, low-risk group; 16–25, intermediate-risk group; 26–100, high-risk group).

### 2.3. Statistical Analysis

For continuous variables, the *t*-test or the Wilcoxon rank-sum test was used to compare two groups, and analysis of variance or the Kruskal–Wallis test was used to compare three groups according to the results of the normality test. The tumor size was classified into four groups (≤1 cm, 1–2 cm, 2–5 cm, >5 cm), and PR and AR expressions were classified into two groups (negative, positive) based on a cutoff of Allred score 2. Categorical variables were compared among the risk groups using the chi-square test or Fisher’s exact test. Binary and multinomial logistic regression analyses were performed to identify the clinicopathological factors that could predict intermediate- and high-risk groups. Multivariable logistic regression used backward selection that included variables with *p* < 0.1 in the univariable analysis and an elimination criterion of *p* < 0.05. A classification tree was constructed to classify the ODX risk groups using clinicopathological variables. Disease-free survival (DFS) was defined as the time from surgery to the first recurrence, metastasis, or death (whichever occurred first). Survival curves were estimated using the Kaplan–Meier method, and the curves were compared using the log-rank test. A *p*-value < 0.05 was considered statistically significant. All statistical analyses were performed using R (version 4.2.1; R Foundation for Statistical Computing, Vienna, Austria).

### 2.4. Ethical Approval

This study was approved by the Institutional Review Board (IRB) of the National Cancer Center, Korea (IRB No. NCC2022-0197) and was performed in accordance with the principles of the Declaration of Helsinki. The need for informed consent was waived owing to the retrospective nature of the study.

## 3. Results

Of the 655 patients who underwent ODX testing following curative resection during the study period, 547 patients with ER-positive, HER2-negative, and LN-negative breast cancer were included in the analysis (Figure 1).

The characteristics of the participants at baseline are presented in Table 1. The patients had a mean age of 47.5 ± 7.8 years, and 73.7% of them were premenopausal. The median tumor size was 1.7 cm (range 0.3–9.5 cm). Four hundred and forty-nine (82.1%) patients had invasive ductal carcinoma, 52 (9.5%) had invasive lobular carcinoma, and 46 (8.4%) had other histopathological features. Four hundred and ninety-eight (91%) patients were PR-positive, and 276 (95.2%) of 290 patients who underwent immunostaining for AR were AR-positive. Only three patients did not receive adjuvant endocrine therapy; two of these in the high-risk group were lost to follow-up, while the other patient in the low-risk group was planning for pregnancy. Adjuvant chemotherapy was administered to 105 (19.2%) patients, most of whom were in the high-risk group, and radiation therapy was administered to 80.4% of the patients. Regarding the ODX RS, 292 (53.4%) patients showed a RS of 0–15, 188 (34.4%) showed a RS of 16–25, and 67 (12.2%) showed a RS of 26–100. In the total study population, histologic grade, nuclear grade, PR expression, Ki-67, and p53 expression were significantly different among the three ODX risk groups (*p* < 0.0001).

Table 2 shows a comparison of the clinicopathological variables among the ODX risk groups according to age groups. The number of patients aged ≤50 years was 379, of which 209 (55.2%), 127 (33.5%), and 43 (11.4%) were in the low-risk (RS 0–15), intermediate-risk (RS 16–25), and high-risk (RS 26–100) groups, respectively. Histologic grade, nuclear grade, PR positivity, AR positivity, Ki-67, and p53 positivity were significantly different between groups in patients aged ≤50 years. One hundred and sixty-eight patients were aged >50 years, and 14.3% of the patients were in the high-risk (RS 26–100) group. Histologic grade, nuclear grade, Ki-67, and p53 expression were significantly different in patients aged >50 years. The representative pathological images for the low-risk and high-risk groups are shown in Appendix A.

The results of the univariable and multivariable logistic regression analyses are presented in Table 3 and Table 4. In patients aged ≤50 years, multinomial univariable analysis revealed that histologic type, histologic grade, nuclear grade, PR expression, AR expression, Ki-67, and p53 expression were significantly associated with the ODX risk group. In multinomial multivariable analysis, high histologic grade (RS 16–25, odds ratio [OR]: 1.467, 95% confidence interval [CI]: 0.681–3.160, *p* = 0.327; RS 26–100, OR: 8.021, 95% CI: 2.942–21.864, *p* < 0.0001), PR negativity (RS 16–25, OR: 2.732, 95% CI: 0.705–10.594, *p* = 0.146; RS 26–100, OR: 79.673, 95% CI: 17.23–368.423, *p* < 0.0001), high Ki-67 (RS 16–25, OR: 1.026, 95% CI: 1.007–1.044, *p* = 0.006; RS 26–100, OR: 1.086, 95% CI: 1.055–1.117, *p* < 0.0001), and p53 positivity (RS 16–25, OR: 2.635, 95% CI: 1.145–6.064, *p* = 0.023; RS 26–100, OR: 4.260, 95% CI: 1.346–13.487, *p* = 0.014) were significantly associated with intermediate-risk or high-risk groups. In patients aged >50 years, the univariable analysis revealed that a large tumor size, high histologic grade, high nuclear grade, high Ki-67, and p53 positivity were significantly associated with the high-risk group (RS 26–100). Only tumor size, Ki-67, and p53 expression were included in the multivariable analysis, and all these variables were significantly associated with the ODX risk group (tumor size, OR: 3.421, 95% CI: 1.192–9.821, *p* = 0.022; Ki-67, OR: 1.061, 95% CI: 1.025–1.099, *p* = 0.001; p53, OR: 7.330, 95% CI: 2.201–24.411, *p* = 0.001).

A decision tree was constructed to classify the three ODX risk groups for patients aged ≤50 years (Figure 2). Histologic grade was identified as the most explanatory variable for the risk group. In the group with a histologic grade of 1 or 2, the order of PR expression, Ki-67, nuclear grade, and p53 expression was confirmed. In cases with a histologic grade of 3, Ki-67 was used as the classification criterion. All included variables were significant in the logistic regression model.

Additionally, in the survival analyses, the median follow-up period was 62.7 months (range 1.0–122.0 months). Twenty-seven patients showed local recurrence or distant metastasis, and only one patient died of breast cancer during the study period. No statistically significant difference in DFS was observed between the risk groups for all patients (*p* = 0.114) and for patients in both age groups (≤50 years, *p* = 0.072; >50 years, *p* = 0.618) (Figure 3).

The intermediate-risk group (RS 16–25) of patients aged ≤50 years received adjuvant treatment, such as endocrine therapy, chemotherapy, and radiation therapy, according to the clinician’s discretion. All 127 patients in this group received adjuvant endocrine therapy. Adjuvant chemotherapy was administered to 29 (22.8%) patients and radiation therapy was administered to 101 (79.5%) patients. No significant difference in DFS was observed according to whether chemotherapy was administered or not (*p* = 0.886) (Figure 4).

## 4. Discussion

In this study, we evaluated clinicopathological factors that can predict the recurrence risk in early breast cancer as an alternative to the expensive ODX test. These associated factors differed according to patient age. In patients aged ≤50 years, histologic grade, PR expression, Ki-67, and p53 expression were significantly associated with the ODX risk group, and in patients aged >50 years, tumor size, Ki-67, and p53 expression were significantly associated with the ODX risk group.

Our finding that the PR status and Ki-67 were associated with the ODX risk group is consistent with the fact that ER, PR, HER2, and Ki-67 are used when calculating the ODX RS. AR status was significantly associated with the ODX risk group in the univariable logistic regression analysis for patients aged ≤50 years (*p* = 0.047), especially in the high-risk group (OR: 5.75, 95% CI: 1.397–23.666, *p* = 0.015). This is because AR is commonly expressed in ER-positive breast cancer at a rate of approximately 70–95%, indicating a relationship between the two hormone receptors [21,22,23]. The tumor size and histologic grade are closely associated with the prognosis of breast cancer. The Nottingham Prognostic Index, which is calculated using the tumor size, number of involved LNs, and tumor grade, is clinically used to determine the prognosis of breast cancer and has been validated in previous studies [24,25,26]. Other studies have revealed that histologic grade plays an important role in the prognosis of breast cancer, especially ER-positive early breast cancer, and have suggested combining histologic grade with disease stage for a more accurate prediction of the prognosis [27,28,29]. p53 expression has been reported to be associated with a shorter DFS and overall survival in patients with triple-negative breast cancer or visceral metastasis [30,31], although data regarding this association were insufficient for our study population. In addition to the pathological factors analyzed in this study, other biomarkers predicting the prognosis of breast cancer have been investigated. Despite differences in study cohorts, previous studies have reported the prognostic value of biomarkers such as BCL2, tumor-infiltrating lymphocytes, specific microRNAs, and circulating tumor cells [32,33,34,35]. Based on these findings, further studies involving patients with early breast cancer are warranted to apply these biomarkers in clinical practice.

Previous studies have also predicted the ODX RS using clinicopathological factors that can be easily obtained from medical records or basic pathological reports [36,37,38,39,40,41,42,43,44]. Orucevic et al. [36] developed a nomogram using six clinicopathological variables, including age, tumor size, tumor grade, PR status, LVI, and histologic type, based on the National Cancer Database. Kim et al. [45] verified whether this nomogram could accurately predict the ODX RS in Korean patients: the concordance index was much lower than that reported in the original study at the University of Tennessee Medical Center (0.642 vs. 0.890 for the commercial ODX cutoff and 0.605 vs. 0.852 for the TAILORx ODX cutoff).

Several studies have been conducted on Korean patients to obtain more accurate results for the Korean population. Lee et al. [37] reviewed 2815 patients and found that ER, PR, nuclear grade, LVI, and Ki-67 were independent predictors of a low-risk ODX RS. Yoo et al. [38] reported that high nuclear grade, no PR expression, and high Ki-67 were significantly associated with high ODX RS. The absence of PR expression and high Ki-67 were identified as common factors in the present study. Other Western studies have shown the PR status and tumor grade to be strong predictors of ODX RS; these were also revealed as associated factors in the present study [36,39,40,41,42]. However, it is notable that, unlike in previous Korean studies, age was identified as a significant factor in some studies [36,46]. In addition, TAILORx reported some benefits of adjuvant chemotherapy in patients aged ≤50 years with a RS of 16–25 [13]. Therefore, we stratified the patients in the present study using a cutoff age of 50 years to demonstrate the age-based differences; the study results showed such differences in the associated factors.

In the present study, the risk group stratification by ODX RS was based on previous studies and a guideline. The cutoff value for the low-risk group in this study differed from that reported in previous studies [36,38]. These studies used the TAILORx cutoff, which classifies ODX RS 0–10 as low risk, 11–25 as intermediate risk, and 26–100 as high risk [11]. However, a recent subgroup analysis by Sparano et al. [13] revealed significant differences in the chemotherapeutic effects of invasive DFS in subgroups of combined age and RS (*p* = 0.004). In women aged ≤50 years, some benefits of adjuvant chemotherapy were realized when the RS was 16–20 (percentage-point difference, 0.8 at 5 years and 1.6 at 9 years) or 20–25 (percentage-point difference, 3.2 at 5 years and 6.5 at 9 years). Therefore, we applied these RS cutoff values and classified patients aged ≤50 years into three risk groups (0–15, low-risk; 16–25, intermediate-risk; 26–100, high-risk). In addition, we performed a survival analysis to evaluate the benefit of chemotherapy in the intermediate-risk group; however, no significant difference in DFS was observed according to whether chemotherapy was administered during the follow-up period. Additional studies are needed to confirm the benefits of chemotherapy in patients aged ≤50 years with a RS of 16–25.

Unlike in previous studies, the data in this study were analyzed by dividing the patients according to a 50-year age cutoff, and some age-based differences in the ODX risk group-associated factors were observed. Therefore, the accuracy of predicting the ODX risk group can be improved by using different pathological factors according to the patient’s age. However, this study has several limitations. First, it was a retrospective study; therefore, a selection bias was present. The distribution of each risk group was also different; in particular, the proportion of patients in the high-risk group was low, with 11.4% of those aged ≤50 years and 14.3% of those aged >50 years. This is because, in this institution, the ODX test is performed in relatively low-risk patients based on the clinician’s recommendation to select patients who can avoid adjuvant chemotherapy. Large-scale studies with more cases in Koreans or Asians are needed to validate these results, especially for predicting a high-risk group. Second, the follow-up period for patients in this study varied from 0.99 months to 122.01 months because the study period was approximately 10 years long and included recent years. Because this study was not designed for a survival analysis, the follow-up period was relatively short. Further studies with long-term follow-up are needed to accurately compare the survival rates. Finally, the rate of contribution of each associated factor in ODX RS prediction was not assessed. Therefore, the ODX RS cannot be accurately predicted using the results of this study alone.

## 5. Conclusions

Clinicopathological factors, including the tumor size, histologic grade, PR expression, Ki-67, and p53 expression, can be used to predict the ODX risk group. The factors associated with the ODX risk group differed according to age. Further large-scale prospective studies and the development of nomograms using these associated clinicopathological factors for application in clinical practice are required.

## Figures and Tables

**Figure 1 cancers-15-04451-f001:**
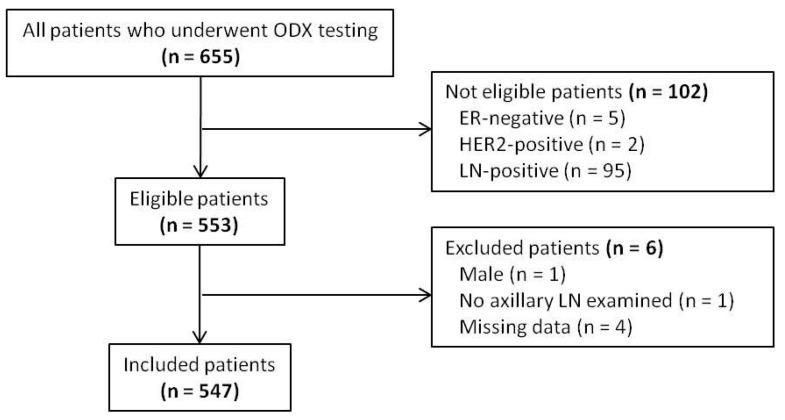
Flow chart showing patient selection. ODX, Oncotype DX; ER, estrogen receptor; HER2, human epidermal growth factor 2; LN, lymph node.

**Figure 2 cancers-15-04451-f002:**
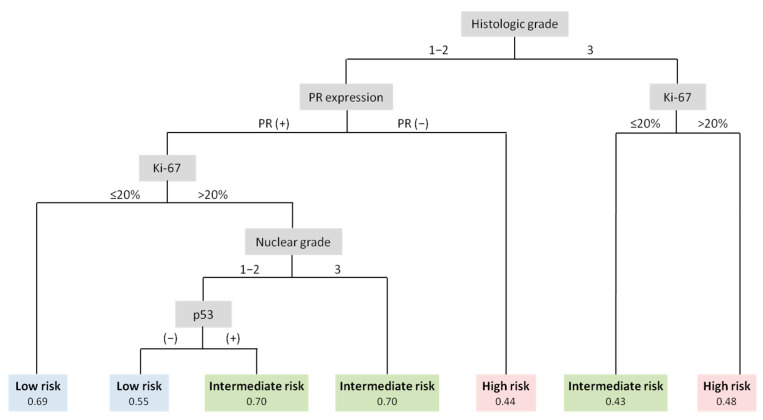
Decision tree for the prediction of the ODX risk group in patients aged ≤50 years. The included variables were histologic grade, PR expression, Ki-67, nuclear grade, and p53, in order. Notes: Patients with histologic grade 1 or 2, PR-positive status, and low Ki-67 had a probability of being in the low-risk group of 0.69. ODX, Oncotype DX; PR, progesterone receptor.

**Figure 3 cancers-15-04451-f003:**
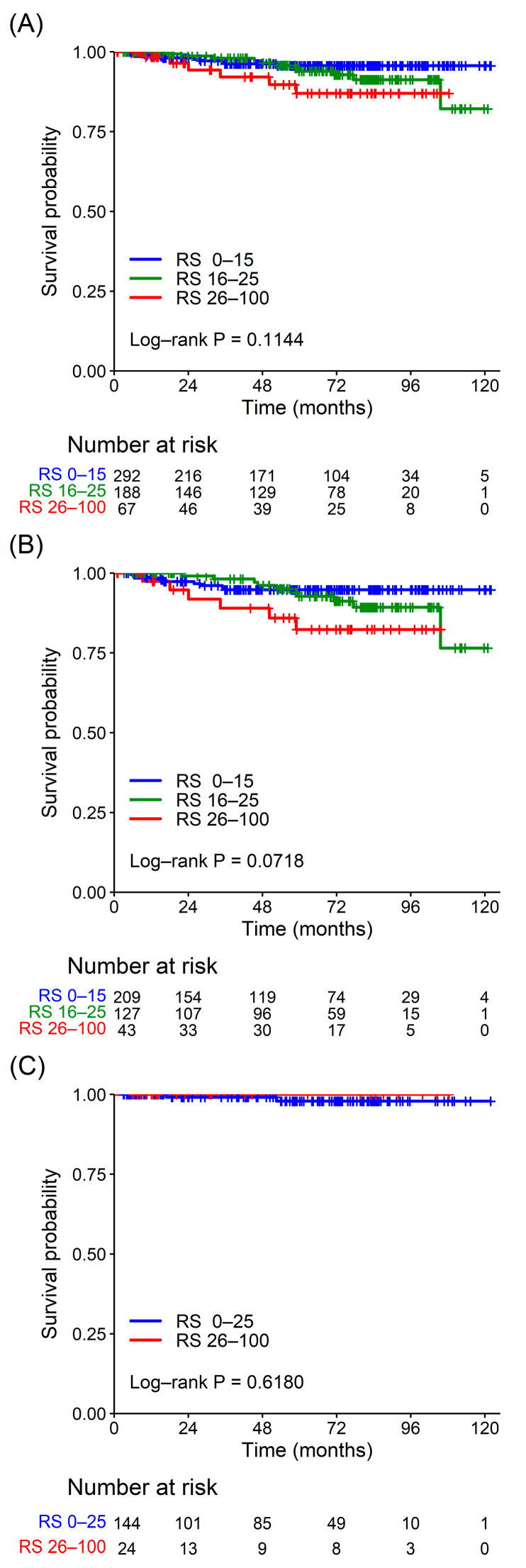
Disease-free survival according to the ODX risk group. (**A**) Disease-free survival in the total study population. (**B**) Disease-free survival in patients aged ≤50 years. (**C**) Disease-free survival in patients aged >50 years. There was no statistically significant difference in disease-free survival between the ODX risk groups in either age group. ODX, Oncotype DX; RS, recurrence score.

**Figure 4 cancers-15-04451-f004:**
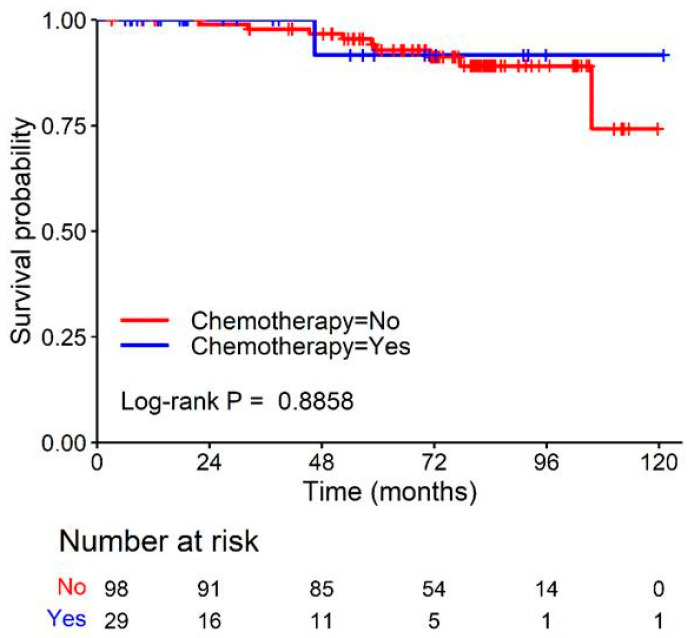
Disease-free survival according to adjuvant chemotherapy in patients aged ≤50 years with an ODX RS of 16–25. There was no significant difference in disease-free survival between patients who underwent chemotherapy and those who did not. ODX, Oncotype DX; RS, recurrence score.

**Table 1 cancers-15-04451-t001:** Baseline characteristics and association between the clinicopathological factors and the Oncotype DX risk group in the study population. Statistically significant results are in bold.

Characteristics	*n* (%)	RS 0–15	RS 16–25	RS 26–100	*p*
*n* (%)	547	292 (53.4%)	188 (34.4%)	67 (12.2%)	-
Age (years)					0.3615
Mean ± SD	47.5 ± 7.8	47.1 ± 7.7	47.7 ± 7.5	48.9 ± 8.8	
≤50	379 (69.3%)	209 (71.6%)	127 (67.6%)	43 (64.2%)	0.4054
>50	168 (30.7%)	83 (28.4%)	61 (32.5%)	24 (35.8%)	
$\mathrm{BMI}\;(\mathrm{kg}/m^{2}$)					0.3904
Mean ± SD	23.3 ± 3.4	23.2 ± 3.2	23.2 ± 3.4	23.8 ± 3.7	
Menopausal status					0.1384
Premenopausal	403 (73.7%)	222 (76%)	138 (73.4%)	43 (64.2%)	
Postmenopausal	144 (26.3%)	70 (24%)	50 (26.6%)	24 (35.8%)	
Tumor size (cm)					
≤1 cm	52 (9.5%)	30 (10.3%)	16 (8.5%)	6 (9%)	0.8158
>1 cm, ≤2 cm	315 (57.6%)	167 (57.2%)	114 (60.6%)	34 (50.8%)	
>2 cm, ≤5 cm	173 (31.6%)	91 (31.2%)	56 (29.8%)	26 (38.8%)	
>5 cm	7 (1.3%)	4 (1.4%)	2 (1.1%)	1 (1.5%)	
Histologic type					0.0842
IDC	449 (82.1%)	230 (78.8%)	158 (84%)	61 (91%)	
ILC	52 (9.5%)	31 (10.6%)	19 (10.1%)	2 (3%)	
others	46 (8.4%)	31 (10.6%)	11 (5.9%)	4 (6%)	
Histologic grade					**<0.0001**
1	76 (13.9%)	50 (17.1%)	22 (11.7%)	4 (6%)	
2	398 (72.8%)	223 (76.4%)	142 (75.5%)	33 (49.3%)	
3	73 (13.4%)	19 (6.5%)	24 (12.8%)	30 (44.8%)	
Nuclear grade					**<0.0001**
Low	12 (2.2%)	10 (2.8%)	1 (0.8%)	1 (1.5%)	
Intermediate	447 (81.7%)	308 (87.3%)	103 (81.1%)	36 (53.7%)	
High	88 (16.1%)	35 (9.9%)	23 (18.1%)	30 (44.8%)	
LVI					0.6422
Negative	319 (58.4%)	176 (60.3%)	106 (56.4%)	37 (56.1%)	
Positive	227 (41.6%)	116 (39.7%)	82 (43.6%)	29 (43.9%)	
PR expression					
Negative	49 (9%)	12 (4.1%)	19 (10.1%)	18 (26.9%)	**<0.0001**
Positive	498 (91%)	280 (95.9%)	169 (89.9%)	49 (73.1%)	
AR expression (*n* = 290)					
Negative	14 (4.8%)	5 (3%)	5 (5.7%)	4 (11.4%)	0.0848
Positive	276 (95.2%)	162 (97%)	83 (94.3%)	31 (88.6%)	
Ki-67 (%)					**<0.0001**
Median (min-max)	16 (1–87)	13.5 (1–76)	17 (1–67)	27 (1–87)	
p53 expression					**<0.0001**
Negative	482 (88.1%)	278 (95.2%)	160 (85.1%)	44 (65.7%)	
Positive	60 (11%)	11 (3.8%)	27 (14.4%)	22 (32.8%)	
Unknown	5 (0.9%)	3 (1%)	1 (0.5%)	1 (1.5%)	
Type of surgery					0.1526
BCS	440 (80.4%)	226 (77.4%)	157 (83.5%)	57 (85.1%)	
Mastectomy	107 (19.6%)	66 (22.6%)	31 (16.5%)	10 (14.9%)	
Anti-hormonal therapy					0.0729
No	3 (0.6%)	1 (0.3%)	0 (0%)	2 (3%)	
Yes	544 (99.5%)	291 (99.7%)	188 (100%)	65 (97%)	
Chemotherapy					**<0.0001**
No	442 (80.8%)	288 (98.6%)	147 (78.2%)	7 (10.5%)	
Yes	105 (19.2%)	4 (1.4%)	41 (21.8%)	60 (89.6%)	
Radiation therapy					0.7773
No	107 (19.6%)	62 (21.2%)	34 (18.1%)	11 (16.4%)	
Yes	440 (80.4%)	230 (78.8%)	154 (81.9%)	56 (83.6%)	

RS, recurrence score; SD, standard deviation; BMI, body mass index; IDC, invasive ductal carcinoma; ILC, invasive lobular carcinoma; LVI, lymphovascular invasion; PR, progesterone receptor; AR, androgen receptor; BCS, breast-conserving surgery.

**Table 2 cancers-15-04451-t002:** Association between the clinicopathological factors and the Oncotype DX risk groups stratified by age. Statistically significant results are in bold.

Characteristics	Age ≤ 50 Years				Age > 50 Years		
	RS 0–15	RS 16–25	RS 26–100	*p*	RS 0–25	RS 26–100	*p*
*n* (%)	209 (55.2%)	127 (33.5%)	43 (11.4%)		144 (85.7%)	24 (14.3%)	
BMI (kg/$m^{2}$)				0.5409			0.5179
Mean ± SD	22.9 ± 3.1	22.7 ± 3.1	23.3 ± 3.7		24.2 ± 3.6	24.7 ± 3.4	
Menopausal status				0.7644			0.1332
Premenopausal	199 (95.2%)	119 (93.7%)	40 (93%)		42 (29.2%)	3 (12.5%)	
Postmenopausal	10 (4.8%)	8 (6.3%)	3 (7%)		102 (70.8%)	21 (87.5%)	
Tumor size (cm)							
≤1 cm	25 (12%)	12 (9.5%)	5 (11.6%)	0.9282	9 (6.3%)	1 (4.2%)	0.0967
>1 cm, ≤2 cm	122 (58.4%)	71 (55.9%)	24 (55.8%)		88 (61.1%)	10 (41.7%)	
>2 cm, ≤5 cm	59 (28.2%)	42 (33.1%)	14 (32.6%)		46 (31.9%)	12 (50%)	
>5 cm	3 (1.4%)	2 (1.6%)	0 (0%)		1 (0.7%)	1 (4.2%)	
Histologic type				0.0891			0.6442
IDC	162 (77.5%)	104 (81.9%)	41 (95.4%)		122 (84.7%)	20 (83.3%)	
ILC	24 (11.5%)	14 (11%)	1 (2.3%)		12 (8.3%)	1 (4.2%)	
Others	23 (11%)	9 (7.1%)	1 (2.3%)		10 (6.9%)	3 (12.5%)	
Histologic grade				**<0.0001**			**0.0055**
1	39 (18.7%)	14 (11%)	3 (7%)		19 (13.2%)	1 (4.2%)	
2	155 (74.2%)	95 (74.8%)	17 (39.5%)		115 (79.9%)	16 (66.7%)	
3	15 (7.2%)	18 (14.2%)	23 (53.5%)		10 (6.9%)	7 (29.2%)	
Nuclear grade				**<0.0001**			**0.0021**
Low	6 (2.9%)	1 (0.8%)	1 (2.3%)		4 (2.8%)	0 (0%)	
Intermediate	181 (86.6%)	103 (81.1%)	21 (48.8%)		127 (88.2%)	15 (62.5%)	
High	22 (10.5%)	23 (18.1%)	21 (48.8%)		13 (9.0%)	9 (37.5%)	
LVI	(*n* = 378)			0.8834			0.3711
Negative	124 (59.3%)	72 (56.7%)	25 (59.5%)		86 (59.7%)	12 (50%)	
Positive	85 (40.7%)	55 (43.3%)	17 (40.5%)		58 (40.3%)	12 (50%)	
PR expression							
Negative	4 (1.9%)	5 (3.9%)	12 (27.9%)	**<0.0001**	22 (15.3%)	6 (25%)	0.2431
Positive	205 (98.1%)	122 (96.1%)	31 (72.1%)		122 (84.7%)	18 (75%)	
AR expression	(*n* = 191)				(*n* = 99)		
Negative	5 (4.2%)	3 (5.9%)	4 (20%)	**0.0377**	2 (2.4%)	0 (0%)	>0.9999
Positive	115 (95.8%)	48 (94.1%)	16 (80%)		82 (97.6%)	15 (100%)	
Ki-67 (%)							
Median (min-max)	14 (1–76)	18 (1–67)	33 (1–87)	**<0.0001**	13 (1–58)	22 (4–80)	**0.0001**
p53 expression	(*n* = 375)						
Negative	196 (95.2%)	109 (85.8%)	29 (69.1%)	**<0.0001**	133 (93%)	15 (62.5%)	**0.0002**
Positive	10 (4.9%)	18 (14.2%)	13 (31%)		10 (7%)	9 (37.5%)	

RS, recurrence score; BMI, body mass index; SD, standard deviation; IDC, invasive ductal carcinoma; ILC, invasive lobular carcinoma; LVI, lymphovascular invasion; PR, progesterone receptor; AR, androgen receptor.

**Table 3 cancers-15-04451-t003:** Univariable and multivariable multinomial logistic regression model in patients aged ≤50 years. Statistically significant results are in bold.

Characteristics	Univariable (*n* = 379)							Multivariable (*n* = 375)						
	Type 3 Analysis of Effects *p*-Value	RS 16–25 (Ref: RS 0–15)			RS 26–100 (Ref: RS 0–15)			Type 3 Analysis of Effects *p*-Value	RS 16–25 (Ref: RS 0–15)			RS 26–100 (Ref: RS 0–15)		
		OR	95% CI	*p*	OR	95% CI	*p*		OR	95% CI	*p*	OR	95% CI	*p*
BMI (kg/$m^{2}$)	0.5404	0.984	0.917–1.056	0.6597	1.044	0.948–1.149	0.3863							
Menopausal status	0.7660													
Premenopausal		1 (ref)			1 (ref)									
Postmenopausal		1.338	0.514–3.484	0.5512	1.493	0.393–5.667	0.5562							
Tumor size (cm)	0.6306													
≤2 cm		1 (ref)			1 (ref)									
>2 cm		1.257	0.785–2.013	0.3412	1.145	0.566–2.313	0.7067							
Histologic type	**0.0454**													
IDC		1 (ref)			1 (ref)									
non-IDC		0.762	0.437–1.329	0.3388	0.168	0.039–0.721	0.0164							
Histologic grade	**<0.0001**							**0.0002**						
1–2		1 (ref)			1 (ref)				1 (ref)			1 (ref)		
3		2.136	1.035–4.406	0.04	14.873	6.704–32.997	<0.0001		1.467	0.681–3.160	0.3274	8.021	2.942–21.864	<0.0001
Nuclear grade	**<0.0001**													
Low/Intermediate		1 (ref)			1 (ref)									
High		1.879	0.999–3.534	0.0504	8.112	3.857–17.06	<0.0001							
LVI	0.8834													
Negative		1 (ref)			1 (ref)									
Positive		1.114	0.713–1.742	0.6345	0.992	0.505–1.949	0.9814							
PR expression	**<0.0001**							**<0.0001**						
Positive		1 (ref)			1 (ref)				1 (ref)			1 (ref)		
Negative		2.100	0.553–7.971	0.2755	19.838	6.017–65.403	<0.0001		2.732	0.705–10.594	0.146	79.673	17.23–368.423	<0.0001
AR expression	**0.0472**													
Positive		1 (ref)			1 (ref)									
Negative		1.438	0.330–6.255	0.6285	5.749	1.397–23.666	0.0154							
Ki-67 (%)	**<0.0001**	1.026	1.008–1.043	0.0034	1.075	1.052–1.099	<0.0001	**<0.0001**	1.026	1.007–1.044	0.0063	1.086	1.055–1.117	<0.0001
p53 expression	**<0.0001**							**0.0227**						
Negative		1 (ref)			1 (ref)				1 (ref)			1 (ref)		
Positive		3.237	1.443–7.259	0.0044	8.786	3.530–21.871	<0.0001		2.635	1.145–6.064	0.0227	4.260	1.346–13.487	0.0137

RS, recurrence score; ref, reference; OR, odds ratio; CI, confidence interval; BMI, body mass index; IDC, invasive ductal carcinoma; LVI, lymphovascular invasion; PR, progesterone receptor; AR, androgen receptor.

**Table 4 cancers-15-04451-t004:** Univariable and multivariable logistic regression model in patients aged >50 years. Statistically significant results are in bold.

Characteristics	RS > 25 (Ref: RS ≤ 25)					
	Univariable			Multivariable		
	OR	95% CI	*p*	OR	95% CI	*p*
$\mathrm{BMI}\;(\mathrm{kg}/m^{2}$)	1.039	0.927–1.164	0.516			
Menopausal status						
Premenopausal	1 (ref)					
Postmenopausal	2.882	0.816–10.179	0.1002			
Tumor size (cm)						
≤2 cm	1 (ref)			1 (ref)		
>2 cm	2.439	1.017–5.852	**0.0459**	3.421	1.192–9.821	**0.0223**
Histologic type						
IDC	1 (ref)					
non-IDC	1.109	0.346–3.558	0.8618			
Histologic grade						
1–2	1 (ref)					
3	5.518	1.856–16.408	**0.0021**			
Nuclear grade						
Low/Intermediate	1 (ref)					
High	6.046	2.216–16.499	**0.0004**			
LVI						
Negative	1 (ref)					
Positive	1.483	0.623–3.528	0.373			
PR expression						
Positive	1 (ref)					
Negative	1.848	0.66–5.175	0.2421			
AR expression						
Positive	N/A					
Negative						
Ki-67 (%)	1.070	1.035–1.107	**<0.0001**	1.061	1.025–1.099	**0.0008**
p53 expression						
Negative	1 (ref)			1 (ref)		
Positive	7.980	2.801–22.733	**0.0001**	7.33	2.201–24.411	**0.0012**

RS, recurrence score; ref, reference; OR, odds ratio; CI, confidence interval; BMI, body mass index; IDC, invasive ductal carcinoma; LVI, lymphovascular invasion; PR, progesterone receptor; AR, androgen receptor.

## Data Availability

The data presented in this study are available from the corresponding author upon reasonable request. The data are not publicly available due to privacy policies.

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
