# Peer review of "Clinicopathological Factors Associated with Oncotype DX Risk Group in Patients with ER+/HER2- Breast Cancer"

_cancers, 2023, doi:10.3390/cancers15184451_

Round 1
Reviewer 1 Report
The authors discuss clinicopathological findings that can be used to predict risk in breast cancer patients that can be substituted for ODX; the content provides an overview of the relationship between both ODX and clinicopathological information and is clinically applicable.
Minor point:
The authors have done a cutoff for breast cancer patients by age, but would it be useful to do a cutoff by pre- and postmenopause instead? Please discuss.
Reviewer 2 Report
Song et al. present an interesting and well-structured manuscript. The authors present interesting results that support the conclusions. Authors need to address a few questions:
-The title must improve and have a clear translational character.
-The introduction to the state of the art must have a justification of the novelty of the study.
-Authors must calculate the statistical potential and the sample size.
-The figures are little self-explanatory and of low quality.
-The figures legends should be described better.
-Authors must include a figure where representative histological images are included.
-The authors should make a more translational discussion, where the manuscripts doi: 10.3390/ijms24098396 are included. and doi: 10.3390/medicina58060722.
-The use of English grammar must be specifically reviewed.
Extensive editing of English language required.
Round 2
Reviewer 2 Report
The authors have responded satisfactorily to the questions and indications. They have made major improvements. However, the quality of the figures is still poor. Please, the authors should improve this point before the final acceptance.Minor points.